Indicators for the use of robotic labs in basic biomedical research: a literature analysis

Groth Paul pgroth@gmail.com p.groth@elsevier.com
Cox Jessica
Elsevier Labs , Amsterdam , Netherlands
Tullius Thomas
Electronic publication date: 2017 Nov 8
Publication date: 2017
Volume: 5
Electronic Location ID: e3997
Received 2017 Jun 26; Accepted 2017 Oct 16
Copyright: ©2017 Groth and Cox
Copyright year: 2017
Copyright holder: Groth and Cox
License: This is an open access article distributed under the terms of the Creative Commons Attribution License, which permits unrestricted use, distribution, reproduction and adaptation in any medium and for any purpose provided that it is properly attributed. For attribution, the original author(s), title, publication source (PeerJ) and either DOI or URL of the article must be cited.
License URL: https://creativecommons.org/licenses/by/4.0/

Keywords: Robotic labs, Indicators, Text mining, Methods, Literature analysis

Funding: The authors received no funding for this work.

==============================
Robotic labs, in which experiments are carried out entirely by robots, have the potential to provide a reproducible and transparent foundation for performing basic biomedical laboratory experiments. In this article, we investigate whether these labs could be applicable in current experimental practice. We do this by text mining 1,628 papers for occurrences of methods that are supported by commercial robotic labs. Using two different concept recognition tools, we find that 86%–89% of the papers have at least one of these methods. This and our other results provide indications that robotic labs can serve as the foundation for performing many lab-based experiments.

Introduction

The reproducibility of a scientific experiment is an important factor in both its credibility and overall usefulness to a given field. In recent years, there has been an uptick in discussion surrounding scientific reproducibility, and it is increasingly being called into question. For example, Baker (2016) conducted a 2016 survey of 1500 researchers for Nature in which 70% were unable to reproduce their colleague’s experiments. Furthermore, over 50% of the same researchers agreed that there was a significant crisis in reproducibility. While these issues arise in all fields, special attention has been paid to reproducibility in cancer research. Major pharmaceutical companies like Bayer and Amgen have reported the inability to reproduce results in preclinical cancer studies, potentially explaining the failure of several costly oncology trials (Begley & Ellis, 2012).

Munafò et al. (2017) outline several potential threats to reproducible science including p-hacking, publication bias, failure to control for biases, low statistical power in study design, and poor quality control. To address these issues, the Reproducibility Project: Cancer Biology in its reproduction of 50 cancer biology papers, used commercial contract research organizations (CROs) as well as a number of other interventions, such as registered reports (Errington et al., 2014). They argue that CROs provide a better basis for replication as they are both skilled in the expertise area and independent, in turn reducing risk of bias.

Extending this approach to providing an industrialized basis for performing experiments, is the introduction of large amounts of automation into experimental processes. At the forefront of this move towards automation is the introduction of “robotic labs”. These are labs in which the entire experimental process is performed by remotely controlled robots through the cloud (Bates et al., 2016). A pioneering example of this is King’s Robot Scientist (King et al., 2009), which completely encapsulates and connects all the necessary equipment in order to perform microbial batch experiments; only needing to be provided consumables. Companies such as Transcriptic (http://transcriptic.com) and Emerald Cloud Lab (http://emeraldcloudlab.com) are beginning to make this same infrastructure in a commercial form. One can see these robotic labs as an extension and democratization of the existing CRO infrastructure focused even more on automation and the accessibility of these labs through Web portals and Application Programming Interfaces (APIs).

The promise of these labs is that they remove the issues of quality control from individual labs and provide greater transparency in their operation. Additionally, they allow for biomedical experiments to become more like computational experiments where code can be re-executed, interrogated, analyzed and reused. This ability to have a much more detailed computational view is critical for reproducibility as narrative descriptions of methods are known to be inadequate for this task as summarized in Gil & Garijo (2017). This lack of detail is illustrated compellingly in the work on reproducibility maps where it took 280 h to reproduce a single computational experiment in computational biology (Garijo et al., 2013). While there are still challenges to reproducibility even within computational environments (Fokkens et al., 2013), robotic labs potentially remove an important variable around infrastructure. They provide, in essence, a programming language for biomedical research. While this does not address existing reproducibility issues with methods described in the literature, it does provide a foundation for more reproducible descriptions in the future.

While this promise is compelling, a key question is whether robotic labs would be widely applicable to current methods used in biomedical research. This question can be broken down into two parts:

1. does basic lab-based biomedical research reuse and assemble existing methods, or is it primarily focused on the development of new techniques; and;

2. what existing methods are covered by robotic labs?

To answer this question, we use an approach inspired by Vasilevsky et al. (2013) that used text analysis of the literature to identify resources (e.g., cell lines, reagents). Concretely, we automatically extract methods from a corpus of 1,628 open access papers from a range of journals covering basic biomedical research. We identify which of those methods are currently supported by robotic labs. Our results show that that 86%–89% of these papers have some methods that are currently supported by cloud-based robotic labs.1

Materials & Methods

Article corpus construction

Our aim was to construct a meaningfully sized corpus that covered representative papers of basic lab-based biomedical research. Additionally, for reasons of processing efficiency, we selected papers from Elsevier because we had access to the XML versions of the paper in a preprocessed fashion.

To build our corpus, we first selected journals categorized under “Life Sciences” in ScienceDirect (http://sciencedirect.com), specifically those marked under “Biochemistry, Genetics and Molecular Biology”. We then filtered for journals categorized as “Biochemistry”, “Biochemistry, Genetics and Molecular Biology”, “Biophysics”, “Cancer Research”, “Cell Biology”, “Developmental Biology”, “Genetics”, or “Molecular Biology”. This returned a list of 412 journals. We then manually inspected each journal on this list. Journals were excluded if they were comprised of seminars or reviews, were non-English, primarily clinical studies, primarily new methods, population studies or a predecessor to another journal. ISSNs were returned for each title, for a final list of 143 journals. The list of journals selected with their ISSN are available at Groth & Cox (2017).

From these journals, we selected all CC-BY licensed papers. The list of papers and their DOIs are available at Groth & Cox (2017) which includes a script to download the corpus.

This selection procedure for articles is visualized in Fig. 1. J. Cox performed the journal selection (i.e., search strategy).

Figure 1 A PRISMA (Moher et al., 2009) flow chart visualizing the article selection procedure.

Method space definition

To define the space of methods, we relied upon the 2015 edition of the National Library of Medicine’s Medical Subject Headings (MeSH) controlled vocabulary. MeSH provides a number of benefits: one, it provides an independent definition of a set of possible methods; two, it provides a computationally friendly definition covering multiple synonyms for the same method concept that researchers could potentially use. For example, it defines synonyms for Polymerase Chain Reaction such as PCR, Nested PCR, and Anchored Polymerase Chain Reaction. Third, because it is arranged hierarchically, it captures methods at different levels of granularity. For example, a researcher may use PCR but not identify the specific variant like Amplified Fragment Length Polymorphism Analysis. Thus, we took the Investigative Techniques [E05] branch of MeSH as defining the total space of methods. For use in our analysis, we extracted that branch from the Linked Data version of MeSH (https://id.nlm.nih.gov/mesh/) using a SPARQL query. This branch of MeSH contained 1,036 total concepts. The SPARQL query, CSV file of the reformated branch, and a link to the specific linked data version are available in Groth & Cox (2017).

To define what methods could be automated by a robot lab, we built a list of available and soon to be available methods from the Transcriptic and Emerald Cloud Lab websites as of March 10, 2017. This list contained 107 methods. The list was constructed by J Cox and verified by P Groth, J Cox was the final decision maker. We term methods that can be executed within a robotic lab a robotic method. We manually mapped those lists to MeSH concepts from the Investigative Techniques [E05] branch. We were able to map 74 methods to MeSH concepts. During the mapping procedure, we searched the MeSH browser using the exact terms listed on the two websites, and selected the exact match as the leaf node of the tree and all of it’s children terms. We assume that children of a parent are often synonymous terms, and this would broaden our coverage of robotic methods. In some cases, this meant that a particular method was mapped to a more general method type. Our final list of robotic methods mapped to MeSH contains 154 unique concepts. The complete mapping is also available at Groth & Cox (2017).

Those methods that were not mapped to a robotic methodbut were tagged with a MeSH investigative technique are termed a non-robotic method.

Method identification

To identify methods mentioned in the corpus, we employed concept recognition. Concept recognition, often called entity linking in the natural language processing literature, is the process of connecting a term to a unique concept identifier in an ontology or taxonomy (Wu & Tsai, 2012). Dictionary-based annotators are commonly used in biomedical concept recognition because the aim is to often recognize many different types of concepts. While machine learning based annotators work extremely well for recognition of specific concepts, e.g., gene/protein recognition (Mitsumori et al., 2005), they require training data for each different domain. Because our aim was to identify methods from a dictionary (i.e., MeSH), we chose to use a dictionary annotator based approach. Tseytlin et al. (2016) provides an overview and performance comparison of existing annotator based tools and shows that existing annotators perform roughly between 0.4 F1 and 0.6 F1 on the biomedical CRAFT (Verspoor et al., 2012) and ShaRe (Suominen et al., 2013) benchmark corpora.

We selected two concept annotators: MetaMap (Aronson & Lang, 2010) and the Solr Dictionary Annotator (SoDA) (Pal, 2015).

Metamap is a widely used concept annotator provided by the National Library of Medicine. It is designed specifically to work well with the Unified Medical Language System (UMLS) vocabulary (Bodenreider, 2004). MeSH is mapped into UMLS. We used the standard settings for MetaMap but limited the tagging to MeSH. We performed a mapping from the resulting UMLS concept IDs to the Mesh concepts using the UMLS terminological web services.

SoDA is an open source flexible, scalable lexicon based annotator that provides convenient integrations with Apache Spark (http://spark.apache.org). (a distributed computing environment). We used SoDA’s lower setting, which searches for the exact terms in both upper and lower case.2

We annotated all content paragraphs (excluding abstracts, titles, section headings, figures, and references) against the whole of MeSH 2015 using both annotators. After annotation, analysis was performed by matching the lists detailed in the previous section with the output annotations. The analysis procedure code is available in Groth & Cox (2017). We used the same code for analysis for the results obtained from both annotators.

Results

Table 1 presents basic statistics of the two annotators presented side by side. Within our 1,628 article corpus, SoDA and MetaMap yielded comparable coverage, returning 1,601 and 1,627 articles with one recognized method, respectively. In total, SoDA returned 387 and MetaMap returned 577 distinct methods. The discrepancies in these numbers reflect the differences in the string matching algorithm utilized by the two concept annotators.

Table 1 Comparison of SoDA and MetaMap Results.

	SoDA	MetaMap	
Articles with one recognized method	1,601	1,627	
Articles without a recognized method	27	1	
Distinct methods tagged	387	577	
Articles with at least one robotic method	1,404	1,454	
Mean number of robotic methods per paper	3.8	4.6	

Table 2 Occurrence of robotic methods.

Method name	SoDA count	MetaMap count	
Polymerase Chain Reaction	720	662	
Transfection	330	337	
Centrifugation	324	324	
Cell Culture Techniques	295	343	
Microscopy	273	491	
Blotting, Western	261	404	
Flow Cytometry	247	272	
Microscopy, Electron, Scanning Transmission	238	1	
Immunoprecipitation	208	221	
Real-Time Polymerase Chain Reaction	194	468	

Using the SoDA mapping, we identified 1,404 articles or roughly 86% of the total corpus to have at least one known robotic method. Of the 1,601 articles with a detected method, the mean number of robotic methods within an article is 3.8. MetaMap identified 1,454 articles, or 89% of the corpus, to have one known robotic method. MetaMap identified 1,627 articles with a method, and a mean number of 4.6 robotic methods within a paper.

Table 2 lists the top 10 most frequently occuring distinct robotic methods, sorted on SoDA count. Of the 74 potential robotic methods, all occurred within our corpus. We analyze this list in more detail later in the discussion section.

Additionally, as discussed we identified the most common non-robotic methods. There were 291 unique non-robotic methods in total, and the 10 most frequently occurring methods, sorted on SoDA count, are presented in Table 3.

Table 3 Occurrence of non-robotic methods.

Method name	SoDA count	MetaMap count	
Observation	695	1,001	
Mass Spectrometry	277	290	
Cell Count	188	204	
Immunohistochemistry	174	189	
Electrophoresis	172	202	
Data Collection	151	175	
Body Weight	135	342	
Immunblotting	134	191	
In Situ Hybridization	121	98	
Mortality	119	122	

Figures 2 and 3 show the overall distribution of the number of unique robotic methods or non-robotic methods detected in each paper, categorized by concept annotator. Figure 2 shows a comparable trend between SoDA and MetaMap, both with a right-skewed distribution, with a large portion of papers having 3–5 robotic methods detected per paper. Figure 3 shows a similar trend, however SoDA found the majority of papers to have a 2–4 unique non-robotic methods per paper while MetaMap yielded the majority to have 7–10 unique non-robotic methods per paper. The difference in these numbers reflects the difference in total detected methods by the two annotators, with MetaMap recognizing more methods overall (577 vs. 387) in a greater selection of papers (1,627 vs. 1,601).

Figure 2 The distribution of the count of unique robot methods per paper, categorized by concept tagger.

Figure 3 The distribution of the count of unique non-robot methods per paper, categorized by concept tagger.

We also measured the total percentage of all detected methods within an individual document that were categorized as robotic methods using both SoDA and MetaMap. Table 4 shows the percentage of documents that had greater than 50%, 75% or total coverage at 100%. SoDA found that 56% of the corpus had at least half of their methods categorized as robotic methods vs. 11% detected by MetaMap. SoDA detected 15% had more than 75% covered, and 3% had complete overlap with robotic methods. MetaMap found 0.2% had at least 75% of their methods covered, and less than 0.05% had complete coverage. Differences in these numbers between annotators is another reflection of their varied detection methods.

Table 4 Percentage of Robot-Methods within a paper.

Percent of all detected methods that are Robot Methods	SoDA frequency	MetaMap frequency	
100%	3%	<0.05%	
≤75%	15%	0.2%	
≤50%	56%	11%	

Discussion

We return to our initial question: (1) do basic biomedical papers reuse existing methods and, (2) if so, are those methods supported by robotic labs.

With respect to the first part of the question, our analysis suggests that biomedical research papers do reuse existing methods. Between 86%–89% of the papers had at least one known method as listed within MeSH. Interestingly, of the potential 1,036 methods 386 were recognized by SoDA and 577 were recognized by MetaMap. Though neither concept annotator recognized 100% of all potential methods, we believe this could be for several reasons. It may be due to the corpus selected, in which these papers employ a smaller number of highly common methods relative to the entire pool. Further, there may be differences in the granularity of reporting methods by scientists within these papers. It is likely also attributed to differences in how the concept annotators work, as well as the coverage of method synonyms in MeSH. From a more qualitative perspective, we see that common techniques are recognized. For example, it is unsurprising that the most common robotic method is PCR, shown in Table 2, and at comparable quantities between the two annotators (SoDA: 720, MetaMap: 662). PCR is a relatively standardized and cost-effective method used ubiquitously in biomedical research. It is an elegant yet straightforward protocol that lends itself to be used in a variety of contexts within a biomedical lab, from gene expression measurement to cloning. Current thermocycler technology enables easy adjustment of experimental parameters, relatively little sample handling and the use of commercialized master mixes. Combined with its pervasiveness in biomedical research labs, these factors make PCR an attractive choice for automation.

Beyond PCR, the other methods in Table 2 are also comprised of highly automatable tasks. Just as thermocycler technology is relatively standardized, so too are the equipment, kits and protocols used for methods like microscopy and Western blotting. Biomedical labs are using nearly identical protocols in many instances, yet introducing their own variability due to human use. In these cases, robotic automation would facilitate quick execution of the same method for all of these labs, increasing transparency and reproducibility. This argument can be extended to all of the methods within the table. Simply stated, robots can pipette, measure and handle samples better than humans can, and in turn facilitate reproducible science.

Table 3 represents the most commonly identified non-robotic methods. Several of the methods listed can be firmly placed in the non-robotic methods category due to either its vague usage (i.e., observation, mortality) or inability to be automated (i.e., body weight). The other terms that appear on this list are commonly used biomedical methods, however the language is either such that it does not cross with the list of cloud lab methods or it is not currently available. For example, electrophoresis is a method commonly used in conjunction with PCR and has the potential to be automated, however it does not appear on the cloud labs list as a standalone method. Conversely, immunoblotting is a more general method that encompasses the robotic method Western blotting; only Western blotting is listed as a robotic method. In our approach, we only listed children of each node and not parents, thus explaining why immunoblotting does not appear as a robotic method. This exposes some “leakiness” in our procedure and should be taken into consideration.

Because there is no ‘gold set’ of methods for each paper, we used two concept annotators to gauge their overall performance throughout the experiment. Tables 2 and 3 demonstrate that SoDA and MetaMap detect methods on the same scale, and the results from both annotators support our general conclusions. To further test performance, we pulled the PubMed indexed terms for each document to test if they could act as a standard for our annotations. However, after crossing the indexed terms with the terms tagged by MeSH or SoDA, we observed that only six papers had more than 50% overlap between these lists, and only 16% of the corpus had any overlap at all. Pubmed indexed terms are meant to act as keyphrases for the document and cannot be expected to capture all methods used within an article. Based on this, we found PubMed indexed terms to be an insufficient source of “standards” and continued using the two concept annotators in parallel.

In terms of the second part of the question, our analysis suggests that the research represented by this corpus of literature has the potential for using robotic labs in at a least some aspects of the described experimental processes. Indeed, looking at the coverage of methods found, one sees that nearly 90% of the methods indexed have some automated equivalent. This figure is striking in that robotic labs are still just becoming available for use, but indicate the potential is great.

Looking more deeply at the actual methods identified, the top robotic methods in Table 2 are a mix of both workflow techniques (i.e., cell culture, transfection) and endpoint measurements (i.e., Western blotting, Real Time PCR). Roughly 3% of our corpus had all of their detected methods supported by cloud labs, which we believe to be an underestimation. This qualitative view provides some support that robotic methods can execute the majority of an end to end biomedical workflow. One may argue that robotic labs do not lend themselves to the building of a disease model. Building a model requires extensive experimentation and parameter tweaking, and some argue that this kind of platform is more conducive to endpoint analysis after a model has been rigorously developed and tested, and not its actual development. However, we contend that with some more work, a robotic lab that does support every part of the workflow would actually accelerate model system development and allow researchers to spend more time developing and testing new hypotheses. This outcome would be the consequence of allowing essentially what is parameter search to be performed by the robot with minimal human interaction during experimental execution. This could accelerate the pace of discovery in entire fields, all while maintaining reproducibility.

While these results provide salient indicators for the ability to move towards robotic labs, there are a number of areas where our analysis could be improved.

Our analysis does not provide information about whether the given automated methods cover all aspects of the protocols described within an article. This incompleteness comes from three sources:

1. the identification relies on a manually created list (i.e., MeSH that is necessarily incomplete);

2. the recognition algorithm does not determine how the methods/steps that are recognized join up to form a total protocol, this includes how materials are physically transferred between steps;

3. papers will frequently not mention steps or smaller parts of protocols that are necessary but are well known to trained researchers.

To address the above, we would need much more complex natural language processing techniques. Indeed, the state of the art in process/task detection (a similar task to method recognition) is only 0.44 F13 that is not including recognizing the dependency relations between the tasks. In biology specific method extraction state of the art ranges between roughly 0.6 and 0.7 F1 (Burns et al., 2016). Recent work by Dasigi et al. (2017) shows the effectiveness of deep learning approaches on the larger scientific discourse extraction; however, this was applied only to a small number of papers. In future work, we aim to apply these recent advances to deepen our analysis. Based on the challenges listed above, we believe that the numbers presented here are an underestimation of the total number of robotic methods that can be applied in biomedical research.

Finally, while we believe the selected corpus reflects the body of literature that would most likely use robotic labs, it could be argued that a much larger corpus would be more informative. This investigation is also left to future work.

Conclusion

Reproducibility is of increasing concern across the sciences. Robotic labs, particularly in biomedicine, provide the potential for reducing the quality control issues between experiments while increasing the transparency of reporting. In this article, we analyzed a subset of the biomedical literature and find that up to 89% of the papers have some methods that are supported by existing commercial robotic labs. Furthermore, we find that basic methods are indeed “popular” and are increasingly being covered by robotic labs.

While there will always be labs that specialize in the development of new methods, given these indicators, we believe that robotic labs can provide the basis for performing a large percentage of basic biomedical research in a reproducible and transparent fashion.

Supplemental Information

Supplemental Information 1 PRISMA checklist

Click here for additional data file.

We thank Ron Daniel, Brad Allen and the reviewers for their helpful comments.

Additional Information and Declarations

Competing Interests

Author Contributions

Data Availability

1 Data and Code are available at http://doi.org/10.17632/gy7bfzcgyd.3 and referenced throughout.

2 As an aside, we found SoDA to be significantly easier to configure than MetaMap within a cloud environment, for example, because MetaMap requires all incoming client IP addresses to be registered.

3 See 2017 SemEval Task 10 https://scienceie.github.io.

Paul Groth and Jessica Cox are employees of Elsevier Labs, Amsterdam, Netherlands.

Paul Groth and Jessica Cox conceived and designed the experiments, performed the experiments, analyzed the data, contributed reagents/materials/analysis tools, wrote the paper, prepared figures and/or tables, reviewed drafts of the paper.

The following information was supplied regarding data availability:

Groth P, Cox J. 2017. Datasets for Potential of Robotic Lab Methods Usage in Biomedical Papers. Mendeley Data, v3 DOI 10.17632/gy7bfzcgyd.3.

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
