# Peer review of "Indicators for the use of robotic labs in basic biomedical research: a literature analysis"

_PeerJ, doi:10.7717/peerj.3997_

## Round 0.1 · original submission · Major Revisions

The three reviewers and I agree that your manuscript addresses an interesting scientific question. However, two of the reviewers (1 and 3) raise important methodological questions that must be addressed in your revision of the manuscript.

Reviewer 1 raises several issues regarding your approach to identifying experimental methods that should be addressed in a revision. I also noted in my reading of your manuscript that a surprisingly high proportion of experimental methods were not able to be identified in MeSH using your approach. This should be discussed in more depth, perhaps based on the suggestions of Reviewer 1.

Of the seven points listed by Reviewer 3, points 4 and 5 also focus on method identification and the general analytical approach that you took. These points need to be addressed in your revision.

Other reviewer comments provide helpful suggestions for editing and improving your manuscript, and you should consider making these changes.

·

Basic reporting

In general, the manuscript is clearly written. There are some typos and small grammatical issues, which should not be difficult to address. I noted the following:
- "methodbut" -> "method but"
- "an robotic" -> "a robotic"
- "methodswithin" -> "methods within"
- "methodsin" -> "methods in"
- "methodscan" -> "methods can"
- "methodsthat" -> "methods that"
- "Finally, While" -> "while"
- "This large overlap could have to do with what the skewed distribution of recognized ..": "what" seems ungrammatical.

Some sentences are unclear or confusing, and I suggest the authors clarify them. These are as follows:
- The authors state that "this ability to have a much more detailed computational view is critical for reproducibility as narrative descriptions of methods are known to be inadequate for this task". I would think the robotic labs would have to start with the narrative description of experiments for replication. How does one go from this narrative description to computational view?
- The authors state that "Of the potential 1035 methods only 151 were recognized". Is 1035 the number of relevant MeSH terms? Earlier in the manuscript, they state there are 1036 relevant MeSH terms.
- " With the exception of cell culture, the other methods in Table 1 are also comprised of highly automatable tasks." I thought being included in Table 1 meant the method could be fully automated. What's the specific difficulty with cell cultures? Explain.
" 26 of these methods are in fact supported by one of these cloud labs, exposing some leakiness in our procedure." The example given is Real-Time PCR. This seems to be a term in MeSH since 2012. Either give a more accurate example or explain the discrepancy.

There are a few figures and tables and their content is adequate. I think Figure 2 can be more effectively presented as a table, and percentages can be added. I think Figure 1 is useful, and something similar for method selection/categorization and for article/method matching would be helpful to guide the reader and tighten up Methods/Results. Currently, these numbers can be hard to follow as narrative text. The authors commendably provide the data and the code for their experiments.

Since method identification (a named entity recognition task) is core to the study, I think it would be useful to provide some background for research in this area. Currently, only towards the end of the Discussion section, this particular task is mentioned and very briefly. A more detailed discussion early in the paper would be more informative and situate this study better.

Experimental design

The notion of automatically reproducing basic research with robots is an intriguing idea that is being increasingly recognized and an investigation of how much of current research can be automated in this manner is timely and meaningful. The authors do a good job of explaining the research goals and providing the context for them.

In general, the methods are described sufficiently and since the data and the code are provided, the methods can be assumed to be replicable.

My main issue with the study is that the method identification approach does not seem powerful enough to make claims about the use of robotic methods. The authors admit this to some extent, stating that their results are likely to underestimate the use of such methods; however, I believe they could have done more with existing, off-the-shelf tools. Much information seems to be lost in mapping steps and also in method identification. I think some expert feedback or use of additional resources would be appropriate.

For example, 71 of 107 methods identified are identified in MeSH (which is a good starting point), the rest (36) seems like a significant proportion to just ignore.Why not try to use other vocabularies/ontologies for these 36 or just generate term lists, since the annotator basically performs simple string matching? For example, OBI (ontology of biomedical investigations) comes to mind as a good candidate. Since the paper is mainly interested in assessing the coverage (and not in only automatic method identification), such interventions would be justified.Some further discussion of these 36 methods and how 71 methods identified in MeSH were mapped to 59 concepts would also be helpful (am I right to assume some methods map to the same MeSH concept?).

It is also not clear to me whether the authors tried using MeSH identifiers used for indexing articles in PubMed. I would think this would be the easiest way of answering their core question. Checking a few of the articles they considered, I can see that indexing terms for these articles include some method names. Since these indexing terms are assigned by human indexers, it can be considered a gold standard and would potentially alleviate the coverage issues noted for the annotation method the authors used. Do the authors find that these methods are not consistently or fully indexed? If the authors have not considered this, I would strongly suggest that they do. If they did, some explanation is needed regarding why they chose not to report it.

There are existing, broad-coverage biomedical named entity recognition methods such as MetaMap (Aronson and Lang, 2010). Did the authors consider using them, instead of coming up with their own lexicon-based method? They state (in the Discussion section) that state-of-the-art in process/task detection is low (0.44 F1), but they do not provide the performance for their own method, making it difficult to assess whether the analysis can be strengthened by such off-the-shelf NER tools. Since MetaMap can be easily tailored to run against MeSH specifically, I believe it can strengthen their approach and findings without too much additional effort.

Validity of the findings

Due to the issues I discussed above, I believe the findings are suggestive rather than conclusive. The authors highlight the difficulties in using automated methods for their task in the Discussion section. I think some of suggestions (using MeSH indexing terms, MetaMap, etc.) can be considered to address some of these challenges (in particular, points 1 and 2 on lines 189-191). However, I believe this paper is still a step in the right direction for understanding the feasibility of such approaches to reproducibility.

As a side note, I think it would still be useful (with the limitations of the approach in mind) to know how many articles (of 1011) were found to contain only robotic methods. This is never clearly stated.

·

Basic reporting

The basic reporting standards have been met.

Experimental design

I see no issues with the experimental design.

Validity of the findings

Overall:
This paper nicely describes the problem of scientific reproducibility and how this could be better addressed by automating basic lab methods using robotics, where applicable. The authors demonstrate that in the corpus of papers analyzed, the majority of methods used in the experiments were assays that could be performed by robotic labs. The conclusions are supported by the analysis.

One thing that I think is missing from the discussion is how this work can impact the future of biomedical research. More specifically, now that we know that the majority of research could be performed by robotics, what are some potential ways we could implement this? A discussion of the potential issues would be nice to include, such as the issue of the expense of running robotic machines, ie not every lab could afford to purchase an automated robotic PCR machine, for example. However, this could be potentially addressed with increased collaborations with labs that already own these tools, and have optimized the protocols. This paper could also be used to promote funding agencies to support automated protocols using robotics, to further ensure scientific reproducibility.

Additional comments

Minor edits:
Abstract - first line
Should it be Robotic labs (lowercase labs)

Lines 11-12
I suggest writing 'could be applicable' in "In this article, we investigate whether these labs are applicable in current experimental practice."

Introduction- Lines 35-36
In this sentence: "These are labs in which the entire
36 experimental process is performed by robots and available remotely in the cloud (Bates et al., 2016)", what is available remotely in the cloud? The data?

Lines 54-56
State the questions as 1a and 1b, otherwise it looks like you are asking two questions.

Discussion Lines 159-161
"Our annotation procedure does not attempt to generalize: the MeSH ID a method is labeled with is exactly what is returned in a search for the method without traversing the hierarchy of the method tree."
Please reread this sentence, there seem to be some typos here.

Line 166
You are asking only one question, in two parts, so I suggest rephrasing to "In terms of the second part of the question..."

Line 174
Missing a space in methodswithin

Line 175
Missing a space in methodscan

Line 191:
Missing right parenthesis

Line 203
Missing a space in methodsthat

Line 205
"While" should be lowercase

--
Did you find any missing synonyms in MeSH? If so, did you make requests to MeSH for these new synonyms?

--
This link does not resolve, it needs to be reformatted: http://dx.doi.org/http://dx.doi.org/10.17632/gy7bfzcgyd.1
--
Shared datasets:
A more extensive description of the datasets in Mendeley would be useful. Currently, the description of data lists the 4 primary datasets, but a detailed description of all the shared files would be helpful. More detailed metadata in each file would be helpful, including a definition of all the acronyms and abbreviations used.

Reviewer 3 ·

Basic reporting

In their paper entitled “Indicators for the use of Robotic Labs in Biomedical Research: A Literature Analysis,” Groth and Cox argue that the larger issue of reproducibility across life sciences research could be addressed by the use of “cloud labs.” These new laboratories are built to fully automate certain biological research protocols. According to the analysis by Groth and Cox, these newly arrived cloud laboratories can automate over 60% of the protocols in a subset of papers within the Elsevier collection. The authors do not clearly claim that researchers should be using these cloud labs, but they suggest that using them for certain protocols will decrease the number of errors within the field and increase the reliability of scientific research.

While an intriguing concept, the paper as is does not warrant publication. At this point it merely points out that certain laboratory methods can be automation, which is obvious to the reader. To warrant publication, the authors must distinguish methods that can be automated AND effect the scientific conclusion of the paper from methods that merely can be placed on a robot. This would be a worthwhile publication.

Experimental design

Point 1 – Definitions of terms

The authors must and clarify define various words used throughout the paper. These include:

1. "automated"
2. A "task" vs. "pipeline. Where automation is used and helpful, and when something is "automated"
3. A "robot" vs. "robotic lab". Robots doing automation are plentiful and prevalent, while cloud labs are not. It's not clear if robots constitute "automation" the same way robotic labs do.
4. Automation of "technique" vs. "analysis.” For example, 'microscopy' appears as an robotic method, but it is unlikely that all instances of microscopy automated both the image capture and image analysis.

Validity of the findings

Point 2 – Methods that could be automated do not seem to be the ones causing the reproducibly issues.

Before publication, the authors must break down methods that can and can not be automated, compared to whether these methods are likely to impact the results of the specific paper. The authors list methods that could be automated by a cloud lab without any examination or discussion of whether automating these methods would have any impact on the final scientific result. Within these, many methods either provide a binary response, and are quite robust to variation and interpretation, or are merely preparatory and the preparation methods are robust to variation. For example, PCR is a robust technique that is usually used in a preparative manner, to obtain enough genetic material for a given experiment. It is difficult to see how outsourcing this method to a CRO would in any way affect the outcome of the primary and important experiment. The same thing could be said for many of the other methods the authors define as automated. A stronger paper would refine further and remove these methods from the list of those that could be automated. Then, a better discussion forms around the methods that can and can not be automated.


Point 3 – How is a fully automated CRO any different than a “cloud lab?”

Before publication, the authors must draw a distinction between the existing, fully automated CROs and a “cloud lab.” Otherwise, the rest of the paper does not come from a position of strength. The authors make a very good point that automation and use of CROs can increase the quality of lab research, but they take a naïve approach to the topic. They are correct that fully automated CROs, or fully automated internal facilities, are the standard for industrial science, where reproducibility is paramount. Where they fail to convince a reader is how these new “Cloud labs” are any different than existing CROs. For example, Transcriptic is a fully automated laboratory that allows researchers to conduct protocols of their choosing. In industry, however, it is widely accepted that a CRO is a fully automated laboratory that allows researchers to conduct protocols of their choosing.


Point 4 -- Analysis lacks depth.

It appears the analysis is a simple word count with little post processing. There is inadequate meaningful justification of this simple technique, or explanation of how the analysis goes further. For example, why was a technique selected that lacked recall? Why were only exact matches used? Why not use a hierarchical ontology for categorizing methods. Additionally, there appear to be substantial error rates in the characterization of technique usage.

Finally, there is insufficient characterization of the papers which did not list an automated method (38% of papers), or discussion of this metric. It is loosely suggested, from the introduction, that these papers are likely to publish a new technique, but this number seems too high.


Point 5 – There is an inadequate characterization of corpus

There is some statement about types of papers in the introduction (main question, part I), but types are never meaningfully characterized initially or analyzed sufficiently. Before publication, the authors should better describe the corpus used for analysis. More metrics, such as the number of papers which have a methods section, would illuminate possible skews in the analysis. The analysis would also benefit from a deeper survey of the types of papers included in the corpus, e.g. MCB, study, clinical, new technique, etc.



Point 6 – Various copy errors

There are various copy errors throughout, but they are not listed here, as the paper must change significantly to warrant publication.

Point 7 - Usefulness of Figure 1

Figure 1 shows several steps which really just reduces down to a single screen of selecting papers which appear in Elsevier.

Additional comments

None

---

## Round 0.2 · accepted · Accept

Please make sure that the typos noted by two of the reviewers are corrected before publication.

·

Basic reporting

The paper is generally well-written. A few typos I pointed out in the previous review still appear in the manuscript, although the authors say that they have been fixed. It might be a PDF conversion error, perhaps. The typos in the current revision are as follows:
- "-The list of papers and their DOIs is available" -> are available
-Typos involving"method" ("methodbut", "methodswithin", etc.)
- "It is an elegant yet straightforward protocol lends itself"-> "that lends itself"
- "due to it's vague usage" -> "its vague usage"

I also think it would be good to be consistent about the usage of the term robotic lab vs. robot lab.

Experimental design

I think the manuscript has been improved with the additional concept annotator and a more inclusive search strategy.
One remaining issue for me: I am not convinced that using simple string matching for the 34 (=107-74) unmatched methods would cause potential bias, as the authors argue in their response. These have already been identified as robotic methods by some external entity, as I understand, not by the authors. Therefore, I believe their use is justified, as it mainly addresses the shortcomings of MESH and would strengthen their findings.

It would also be good to give a sense of how accurate the concept annotators are on a benchmark corpus, if possible, in the lack of a ground truth for the task.

Validity of the findings

No comment.

·

Basic reporting

There are several instances where two words are missing a space, for example:
Page 6/Line 168 and 170 methodsdetected should be methods detected
Page 6/Line 171 and 172 methodsper should be methods per

Otherwise, everything looks good, the paper is well written and referenced, and it meets all the standards for PeerJ.

Experimental design

No comment.

Validity of the findings

No comment.

Additional comments

The only other minor issue is this URL still does not resolve for me:
http://dx.doi.org/http://dx.doi.org/10.17632/gy7bfzcgyd.1

This URL does work:
http://dx.doi.org/10.17632/gy7bfzcgyd.1

Reviewer 3 ·

Basic reporting

See final noted.

Experimental design

See final notes.

Validity of the findings

See final notes.

Additional comments

The paper is much stronger and clearer now, but the analysis still leaves something to be desired. If you look back on my initial concerns, they focused on whether automation of the tasks that the authors suggest, such as PCR, would have any major impact on the outcome of the research. My opinion is that they would not. The authors do not address this claim in their responses. That is the major issue I still have with the manuscript.

I am not firmly against publishing the work, but do believe the tone of the paper needs to change. After all, this is really a presentation of the workflows that are used in the literature vs the availability of these workflows at CROs or automated labs. It is a survey. Conjecturing that automation of these workflows will increase reliability in the scientific realm is not believable to me, and may not be for the reader either. An honest and thorough address of the topics in my original review would flush a lot of this out. Unfortunately, this did not occur.